# 2-Oxonanonoidal Antibiotic Actinolactomycin Inhibits Cancer Progression by Suppressing HIF-1α

**DOI:** 10.3390/cells8050439

**Published:** 2019-05-10

**Authors:** Jiadong Cheng, Lan Hu, Zheng Yang, Caixia Suo, Yueyang Jack Wang, Ping Gao, Chengbin Cui, Linchong Sun

**Affiliations:** 1Division of Molecular Medicine, Hefei National Laboratory for Physical Sciences at Microscale, the CAS Key Laboratory of Innate Immunity and Chronic Disease, School of Life Sciences and Medical Center, University of Science and Technology of China, Hefei 230027, China; jdcheng@mail.ustc.edu.cn (J.C.); hulan@mail.ustc.edu.cn (L.H.); lqyinxue@mail.ustc.edu.cn (Z.Y.); pgao2@ustc.edu.cn (P.G.); 2Guangzhou First People’s Hospital, School of Medicine and Institutes for Life Sciences, South China University of Technology, Guangzhou 510006, China; cxsuo@sibs.ac.cn (C.S.); wangyy.jack@outlook.com (Y.J.W.); 3Third Institute of Oceanography, Ministry of Natural Resources, Xiamen 361005, China

**Keywords:** ALM, hypoxia, HIF-1α, inhibitor, mTOR, combination therapy

## Abstract

HIF-1 serves as an important regulator in cell response to hypoxia. Due to its key role in promoting tumor survival and progression under hypoxia, HIF-1 has become a promising target of cancer therapy. Thus far, several HIF-1 inhibitors have been identified, most of which are from synthesized chemical compounds. Here, we report that ALM (Actino**L**actoMycin**)**, a compound extracted from metabolites of *Streptomyces flavoretus*, exhibits inhibitory effect on HIF-1α. Mechanistically, we found that ALM inhibited the translation of HIF-1α protein by suppressing mTOR signaling activity. Treatment with ALM induced cell apoptosis and growth inhibition of cancer cells both in vitro and in vivo in a HIF-1 dependent manner. More interestingly, low dose of ALM treatment enhanced the anti-tumor effect of Everolimus, an inhibitor of mTOR, suggesting its potential use in combination therapy of tumors, especially solid tumor patients. Thus, we identified a novel HIF-1α inhibitor from the metabolites of *Streptomyces flavoretus,* which shows promising anti-cancer potential.

## 1. Introduction

Hypoxia is one of the most important characteristics of solid tumor microenvironment as a result of cancer cells’ malignant proliferation and lacking of blood supply caused by abnormal structure of vessels in tumor [1,2]. Tumor cells suffer from the lack of O_2_, nutritional deficiencies and low pH under hypoxia, while hypoxia-inducible factor 1 (HIF-1) serves as the major player in cell survival in that case [3,4]. As a transcription factor, HIF-1 regulates hundreds of genes involved in invasion/metastasis, vascularization, genetic instability and treatment failure, making it critical in cancer pathogenesis [5,6,7,8]. Recent studies also revealed that activation of HIF-1 pathway reprogrammed cancer metabolisms including glycolysis, amino acid and lipid metabolism [9,10,11,12,13,14]. Clinical results showed that HIF-1 is overexpressed in most solid tumors and positively correlated with poor prognosis in cancer patients. Therefore, targeting HIF-1 became a promising strategy to treat cancer [15,16,17,18,19].

HIF-1 is a heterodimer composed of O_2_-dependent HIF-1α and constitutively expressed HIF-1β [20,21]. Via proline hydroxylase domain protein 2 (PHD2), O_2_ hydroxylates 402 and 564 proline residues in HIF-1α oxygen-dependent degradation (ODD) domain required for proteasomal degradation, leading to low expression of HIF-1α protein and suppression of HIF-1 pathway activity under normoxic condition [22,23]. In addition to O_2_, HIF-1α is also regulated by upstream signaling pathways, which include the mTOR-p70-RPS6 axis. Mammalian target of rapamycin (mTOR) is a highly conserved serine-threonine protein kinase that regulates ribosomal biogenesis and protein synthesis by phosphorylating P70 ribosomal protein S6 kinase 1 (P70-S6K1) and the eukaryotic initiation factor 4E (eIF-4E)-binding proteins (4E-BPs) [24,25]. Activation of P70-S6K1 leads to increased activity of ribosomal protein S6 (RPS6), which induces translation of HIF-1α mRNA into protein, thus serving as the major determinant of the rate of HIF-1α protein synthesis in many cancers [26].

A growing number of anticancer agents have been shown to inhibit HIF-1 pathway, the mechanism of which includes reduction in HIF-1α mRNA or protein levels, activity of binding to DNA, or HIF-1-mediated transactivation of downstream genes [27]. However, most of these HIF-1 inhibitors were originally approved for targeting other molecules, while their inhibitory effect on HIF-1was recognized through empirical testing later. For this reason, there are no specific HIF-1 inhibitors currently, and developing novel anticancer drugs specifically targeting HIF-1 becomes urgently important.

In our study, we aimed at finding novel HIF-1 inhibitors with inhibitory effects on cancer cell growth and revealing their mechanisms of action. Based on dual luciferase assay, we identified Actino**L**actoMycin (ALM) as a HIF-1 inhibitor, which suppressed cancer cell growth in a hypoxia-dependent manner. Our results further demonstrated that ALM blocked HIF-1α protein synthesis via inhibiting Akt and mTOR signaling. Both pretreatment and delayed treatment of ALM caused growth inhibition of xenograft model in mice.

## 2. Material and Methods

### 2.1. Cell Culture, Cell Proliferation, Apoptosis Assay and Hypoxia Condition

Hep3B cells and PC3 cells were cultured in DMEM and 1640 medium, respectively. The medium was supplemented with 1% penicillin/streptomycin and 10% fetal bovine serum (FBS). For the cell proliferation assay, 5 or 10 × 10^4^ cells were seeded in 12-well plates and counted daily for 4 days. For the apoptosis assay, cells were cultured in indicated conditions, followed by harvesting and Annexin V/PI staining according to standard protocol. Flow-cytometer was used to determine apoptosis population (the first quadrant plus the fourth quadrant). Hypoxia condition was achieved by placing cells in a hypoxic working station (Whitley H35 Hypoxystation, Don Whitley Scientific/DWS), which contains 1% O_2_, 5% CO_2_, and 94% N_2_. The temperature is 37 °C.

### 2.2. Luciferase Assay

p2.1 (in which a hypoxia response element (HRE) from ENO1 gene was inserted upstream of firefly luciferase coding sequence and led to hypoxia-dependent firefly luciferase expression) and pSV-Renilla co-transfected Hep3B cells were seeded in 96-well plates at 1 × 10^4^ cells/well. The following day, the cells were treated with vehicle or ALM at different doses for 24 h. The ratio of firefly/Renilla luciferase was determined by using the Dual Luciferase Assay System (Promega Corp., Madison, WI, USA

### 2.3. Western Blot and Immunoprecipitation

For Western blot, proteins were extracted from cells/tumor samples by using RIPA buffer (50 mM Tris-Cl (pH 8.0), 150 mM NaCl, 5mM EDTA, 1% NP-40, 0.1% SDS). Aliquots of protein samples (100 µg) were resolved by SDS–gel electrophoresis and transferred into nitrocellulose membranes. The membranes were blocked in TBST (10x TBST: 24.2g Tris, 80g NaCl, Tween-20 to 1%, pH to 7.6 and QS to 4L) with 5% milk for 1 h, and then incubated with HIF-1α (BD Biosciences 610959, San Jose, CA, USA), β-actin (Abmart P30002M, Shanghai, China), caspase-3 (Novus Biologicals NB500-210, Briarwood Avenue, CO, USA), RPS6, p-RPS6 (R&D Systems AF3918, Minneapolis, MN, USA), p70, p-p70, mTOR, p-mTOR (2448), p-mTOR (2481), 4E-BP1, p-4E-BP1, Akt and p-Akt (ser473) (Cell Signaling Technology #9202, #9204, #2972, #5536, #2974, #9644, #9451, #4685, and #4060, Danvers, MA, USA). After being washed with TBST for three times, the membranes were incubated with their secondary antibodies. The proteins were visualized with an enhanced chemiluminescence detection reagent.

For immunoprecipitation, Hep3B cell lysate was incubated with 1:10,000 HIF-1α antibody and 40 µL protein A agarose beads for 2 h. The beads were spined down and heated to 100 °C in loading buffer to dissociate HIF-1α.

### 2.4. Pulse-Labeling

Hep3B cells were treated with vehicle (Vhc), 25 nM rapamycin (Rapa), 20 μg/mL cycloheximide (Chx), or 200 nM ALM for 4 h and pulse-labeled by [^35^S] methionine/cysteine for 1 h followed by cell lysis. Aliquots of the cell lysates were afterwards used for autoradiography, Western blot or immunoprecipitation (IP).

### 2.5. Real Time-PCR

Total RNA was extracted from cells/tumor samples by using TRIzol (Life Technologies Corp., Carlsbad, CA, USA). 1 μg of total RNA was treated with DNase, followed by reverse transcription. Real-time PCR was performed by using IQ SYBR Green Supermix and the iCycler Real-Time PCR Detection System (BioRad, Hercules, CA, USA). Expression of HIF-1α, HK1 and Bnip3 relative to 18s was calculated based on the threshold, primer sequences are listed in Appendix A.

### 2.6. Animal Experiment

BALB/C Nude male mice used in this study were performed according to protocols approved by the University of Science and Technology of China (U.S.T.C) Animal Ethics Committee (ethic approval number: USTCACUC1701025). Mice are from Hunan SJA Laboratory Animal Co. Ltd. (Shanghai, China).

PC3 cells were harvested and re-suspended at 2 × 10^7^/mL in a 50:50 mix of PBS: Matrigel (BD Biosciences). 2 × 10^6^ cells were injected into mice subcutaneous tissue on the flank. Tumor volume was calculated as 0.52 × length × width × depth. Vehicle used in animal experiment: 78% saline + 20% corn oil + 1% ethanol + 1% tween-20 (ethanol and tween-20 act as emulsifier). A tumor sample was harvested and divided to two parts for RT-PCR and Western blot.

### 2.7. Statistical Analysis

The data were presented as the mean ± SD or mean ± SEM as indicated. All data are from three independent experiments. Student’s *t*-test was used to calculate *P* values. Statistical significance is displayed by * (*P* < 0.05) unless otherwise noted.

## 3. Results

### 3.1. ALM Inhibits HIF-1α Transactivity and Protein Expression

To identify new HIF-1α inhibitors, hepatocellular carcinoma cells (Hep3B) were stably transfected with p2.1, in which a hypoxia response element (HRE) from ENO1 gene was inserted upstream of firefly luciferase coding sequence and led to hypoxia-dependent firefly luciferase expression, and pSV-Renilla, in which renilla luciferase is consistently expressed. By doing luciferase assay using this established stable cell line, we found that ALM, a kind of 2-oxonanonoidal antibiotic produced during fermentation of the actinomycete strain *Streptomyces flavoretus* isolated from a soil sample from Chinese Yunnan Province, dose-dependently inhibits firefly luciferase expression under hypoxic condition (Figure 1A). To confirm the inhibitory effect of ALM on HIF-1 transcriptional activity, we investigated the effect of ALM on the expression of mRNAs of HIF-1 target genes, such as Bnip3 and HK1 [13,19]. Hep3B and PC3 cells were cultured and treated with different doses of ALM for 24 h under both normoxia and hypoxia, followed by a total RNA isolation and quantitative real-time reverse transcription-PCR (qRT-PCR) analysis. The levels of mRNAs encoding Bnip3 and HK1 decreased dose-dependently in ALM-treated cells, under both normoxia and hypoxia in Hep3B and PC3 cells (Figure 1B,C).

Western blot results revealed that ALM efficiently down-regulates HIF-1α protein expression in Hep3B cells under hypoxic condition in a dose-dependent manner (Figure 1D, upper panel). In human prostate cancer PC3 cells, which have a detectable HIF-1α protein basal level under normoxia (20% O_2_), ALM dose-dependently reduced HIF-1α protein expression under both normoxic and hypoxic conditions (Figure 1D, lower panel). A time course treatment was also conducted. In the presence of 100 nM of ALM, the expression of HIF-1α in PC3, under both normoxia and hypoxia, was completely wiped out after 4 h of drug treatment (Figure 1E). Collectively, these data demonstrated that ALM is a potential HIF-1α inhibitor.

### 3.2. ALM Inhibits HIF-1α Translation by Down-Regulating AKT and mTOR Activity

To investigate the underlying mechanism of ALM inhibition on HIF-1α protein expression, we first checked the effect of ALM on HIF-1α mRNA expression. QRT-PCR revealed that the level of mRNA encoding HIF-1α was not affected by ALM treatment in either Hep3B or PC3 cells, indicating that ALM does not affect transcription of HIF-1α mRNA (Appendix A). Besides hypoxia, HIF-1α can also be induced by the treatment of cobalt chloride (CoCl_2_), desferrioxamine (DFX), or dimethyloxalylglycine (DMOG), each of which is an inhibitor of prolyl hydroxylases (PHDs) that target HIF-1α for VHL-dependent ubiquitination and proteasomal degradation. HIF-1α induced by each of these agents was also blocked by treatment of ALM in both Hep3B and PC3 cells (Figure 2A, upper panel and middle panel). Furthermore, in the presence of MG132, a proteasome inhibitor, ALM still inhibits HIF-1α protein in both Hep3B and PC3 cells (Figure 2A, upper panel and lower panel), indicating that ALM does not inhibit HIF-1α by promoting its PHD-VHL dependent proteasomal degradation.

Since ALM does not inhibit HIF-1α mRNA expression and not promote HIF-1α protein degradation, we hypothesize that ALM might inhibit the translation of HIF-1α mRNA into protein. We pulse-labeled Hep3B cells with [^35^S]-methionine/cysteine, to see if ALM inhibits HIF-1α protein translation in the presence of cycloheximide, a global protein synthesis inhibitor; or rapamycin, an mTOR inhibitor, which also inhibits HIF-1α translation; or 200 nM of ALM. HIF-1α protein was immunoprecipitated from the whole cell lysate and further analyzed by SDS-PAGE followed by autoradiography. Interestingly, similar to rapamycin, ALM efficiently inhibited the synthesis of HIF-1α protein, while the whole protein synthesis was not greatly affected (Figure 2B). As a wide broad protein synthesis inhibitor, cycloheximide thoroughly inhibited whole protein synthesis as well as HIF-1. The results suggested that ALM blocked HIF-1α protein translation.

mTOR is one of the HIF-1α protein translation regulators [26]. The phosphorylation of mTOR is regulated by multiple pathways, among which the PI3K pathway is the most important and frequently studied. mTOR has two primary phosphorylation sites, Akt-regulated Ser2448 and auto-phosphorylation site of Ser2481 [28]. Upon phosphorylation, mTOR activates downstream P70-S6K1, RPS6 and blocks 4E-BP1. In order to investigate the role of mTOR pathway in the inhibitory effect of ALM on HIF-1α, Hep3B cells were treated with vehicle or ALM at different concentrations for 24 h. Western blot results showed that ALM significantly reduced p-Akt at Ser473 (Figure 2C; Appendix A). The phosphorylation of mTOR at Ser2448 was also inhibited, while the auto-phosphorylation site of Ser2481 was not changed (Figure 2C), indicating that ALM suppressed mTOR activity via inhibiting Akt. As for the downstream of mTOR pathway, the phosphorylation of P70, 4E-BP1 and RPS6 were all dramatically inhibited by ALM, without affecting their total protein levels in PC3 as well as Hep3B cells (Figure 2C–E; Appendix A), suggesting that ALM inhibits HIF-1αprotein translation via the mTOR pathway. As expected, the mTOR inhibitor, rapamycin, significantly reduced p-RPS6 and HIF-1α protein (Appendix A).

### 3.3. ALM Induced HIF-1α-Dependent Growth Inhibition and Apoptosis in PC3 Cells

Since HIF-1 plays very important roles in cell growth, proliferation, survival, apoptosis and autophagy in cancer cells, we then studied whether ALM, a potent inhibitor of HIF-1 pathway, affects cancer cell proliferation and apoptosis. PC3 cells were treated with vehicle or 0.1 μM of ALM under both normoxic and hypoxic conditions. Cell growth analysis showed that ALM markedly inhibited PC3 cell growth under hypoxia (Figure 3A), which is consistent with the high expression of HIF-1αprotein under hypoxic condition. We also collected PC3 cell samples for Western blot to detect caspase 3, a 34 kD proenzyme, which is cleaved and activated (17 kD) during apoptosis. As a result, caspase 3 was gradually cleaved in PC3 cells treated with 0.1 μM ALM (Figure 3B). Flow cytometry analysis revealed that ALM significantly induced apoptosis in PC3 cells especially under hypoxia in a time-dependent manner (Figure 3C,D; Appendix A).

To determine if HIF-1α is involved in ALM-suppressed cell proliferation, PC3 cells were stably overexpressing EV, HIF-1α wild type or its P402A/P564A mutants (P402 and p564 are two hydroxylation sites in the oxygen-dependent degradation domain) (Figure 3E). Flow cytometry data showed that in these stable PC3 cell lines with high HIF-1α expression, ALM induced less apoptosis as compared with the empty vector (EV) control group (Figure 3F,G), suggesting that HIF-1α is important for the suppressive effect of ALM in cancer cells. Taken together, these data documented that ALM induced HIF-1α dependent cell apoptosis to suppress cancer cell growth.

### 3.4. ALM Treatment Inhibited PC3 Xenograft Growth

To figure out the effect of ALM on tumor cell growth in vivo, we first used a tumor prevention model. PC3 cells were subcutaneously (s.c.) injected into BALB/c nude mice, which were treated with vehicle or ALM (10 mg/kg body weight) by daily intraperitoneal (i.p.) injection starting from 7 days before tumor inoculation. As a result, ALM significantly inhibited PC3 xenograft growth, with no evident effect on mouse body weight (Figure 4A,B; Appendix A). Western blot analysis using the tumor tissue lysates also confirmed that HIF-1α protein was suppressed by ALM treatment in the tumors (Figure 4C), which is consistent with the in vitro data.

In another independent mouse xenograft experiment, PC3 cells were subcutaneously (s.c.) injected into BALB/c nude mice. 8 days after tumor inoculation, when the tumor volume reached about 200 mm^3^, mice started to be treated with vehicle or ALM (10 mg/kg body weight). In this setting, ALM markedly arrested tumor growth, while the mouse body weight was not affected (Figure 4D–E; Appendix A). Western blot analysis using the lysates from these tumor tissues also showed that suppressed HIF-1α protein by ALM treatment (Figure 4F), confirming that ALM blocked tumor growth by suppressing HIF-1α protein.

### 3.5. Low Dose of ALM Enhances the Inhibitory Effect of mTOR Inhibitors on Cell Growth

Dysregulation of mTOR pathway was observed in multiple tumors, as mTOR regulates translation, protein stability, cell cycle and survival. Since targeting mTOR was demonstrated as a promising strategy against cancer, several mTOR inhibitors (Temsirolimus, Everolimus, etc.) were approved for treatment of multiple cancers, including advanced breast cancer, pancreatic neuroendocrine tumors, advanced renal cell carcinoma and subependymal giant cell astrocytoma [29,30,31,32,33,34]. To determine whether ALM, a novel compound down-regulates AKT and mTOR activity, can improve anti-cancer efficiency of mTOR inhibitors, we performed experiment to evaluate the effect of ALM in the combination therapy with rapamycin (the classic mTOR inhibitor) or Everolimus. Cell proliferation assay showed that low dose Everolimus (10 nM) or rapamycin (30 nM), marginally inhibited cell growth, but low dose of ALM (30 nM) significantly potentiated the inhibitory effect of 10 nM Everolimus or 30 nM rapamycin on cell growth in both PC3 and Hep3B cells, suggesting the very potent inhibitory effect of ALM on cancer cell growth in combination therapy (Figure 5A; Appendix A). More interestingly, PC3 tumor xenograft experiment showed that low dose of ALM (3 mg/kg) significantly enhanced the tumor inhibitory effect of Everolimus at low dose (3 mg/kg) (Figure 5B–C). It should be noted that combination of ALM treatment with Everolimus showed no adverse effect on mouse body weight (Appendix A). Consistent with the tumor growth data, Western blot analysis demonstrated that HIF-1α protein was reduced in xenografts treated by ALM combined with Everolimus (Figure 5D). Taken together, these data documented that ALM enhances the anti-tumor effect of mTOR inhibitors both in vitro and in vivo, suggesting its therapeutic potential in combination therapy.

## 4. Discussion

There has been an intense interest in developing novel therapeutic strategies to target HIF-1α in cancer therapy for the following reasons: (1) HIF-1α expression has been found high in the majority of tumors, especially solid tumor, because of hypoxia, (2) HIF-1 is a master regulator for many aspects in cancer biology, and (3) inhibition of HIF-1α may exploit tumor hypoxia by converting it from a treatment obstacle into a targeting advantage. In this study, we identified one effective HIF-1 inhibitor named ALM. Using Western blot, qRT-PCR, and pulse labeling experiments, we made it clear that ALM inhibits HIF-1α translation (Figure 2). We also confirmed that ALM induces tumor cell apoptosis in vitro and inhibiting tumor xenograft in vivo, by inhibiting HIF-1α (Figure 3 and Figure 4). Mechanistically, we found that ALM inhibits HIF-1α by suppressing mTOR pathway. These findings revealed a novel HIF-1α inhibitor and a potential anti-tumor agent and also indicated the importance and significance of screening new HIF inhibitors from natural compounds. What’s more, combination therapy with ALM significantly enhanced mTOR inhibitors’ effect on tumor suppression. ALM also inhibits HIF-1α protein expression in cells with high basal level of HIF-1α under normoxic conditions as shown in Figure 1D,E which indicates its potential implication in tumors with high levels of HIF-1α. In brief, the results provided evidence for potential clinical application of ALM.

Actinolactomycin, whose structure was revealed as 4,7-dihydroxy-3,9-dimethyl-2-oxonanone, was extracted from fermentation broth of *Streptomyces flavoretus* 18522. Despite its good solubility in DMSO for in vitro use, ALM’s solubility in polar solvents or nonpolar solvents for in vivo study was lower in xenograft experiment due to its emulsion form. Considering the instability of emulsion, chemical modifications need to be employed in order to improve ALM’s solubility or reduce the effective dose for further studies and clinical application. Nowadays, almost 30% of patients received Everolimus exhibits severe side effects, such as mouth sores, stomatitis, infection, mucositis and so on [35,36]. In addition, chronic rapamycin administration results in a diabetes-like syndrome due to hyperlipidemia, glucose intolerance, and reduced fat mass [37]. In our study, we found that low dose of ALM significantly enhanced the tumor inhibitory effect of low dosage Everolimus both in vitro and in vivo without affecting mouse body weight (Figure 5A–C). These results suggest the potential of ALM in combination therapy with mTOR inhibitors for solid tumors. Furthermore, it will be more interesting to explore whether ALM has unknown side effect and special advantages compared with other HIF-1α inhibitors which is not emphasized in this study.

In recent years, more and more natural compound libraries were applied for drug screening and some ones early identified have been employed in clinical trials and application, such as artemisinin for malaria, statins for cardiovascular disease, and taxol for ovarian cancer. Our study provided more evidence of the feasibility to screening anti-tumor drugs from natural compound libraries. Considering the abundance of natural compound and complexity of cancer, more efficient and more specific targetable screening strategies need to be developed.

## Figures and Tables

**Figure 1 cells-08-00439-f001:**
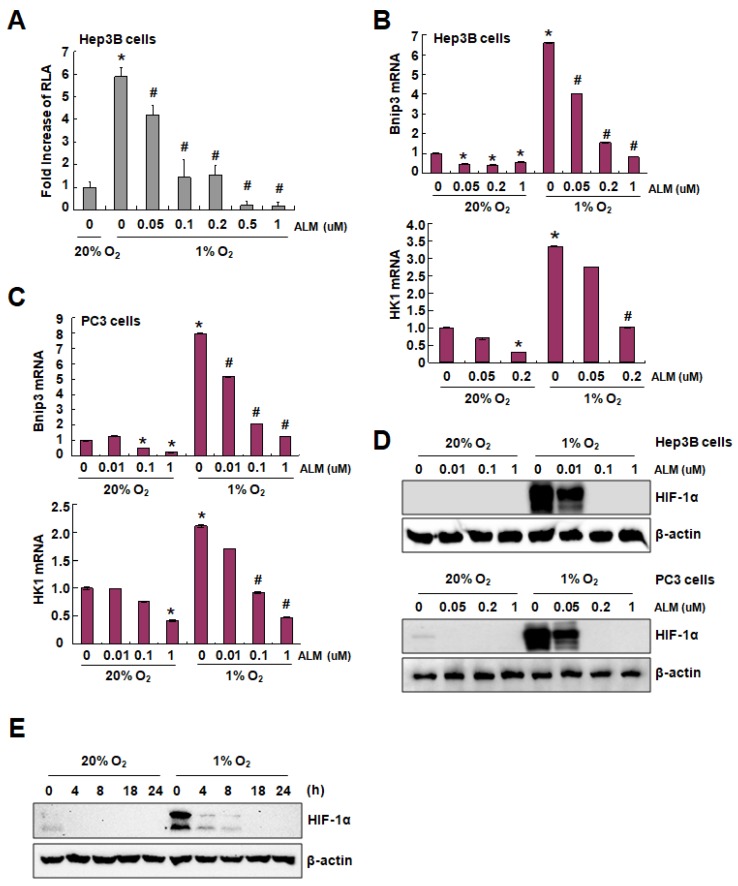
ALM inhibits HIF-1α transactivity and protein expression. (**A**) Hep3B cells stably expressing P2.1 and pSV-Renilla were exposed to normoxic or hypoxic culture conditions and the effect of ALM on the ratio of firefly/Renilla luciferase activity in hypoxic cells was determined; mean ± SD (*n* = 3) are shown. (**B**) Hep3B cells were exposed to vehicle (DMSO) or the indicated concentration of ALM for 24 h under normoxic or hypoxic conditions and total RNA was subjected to RT-PCR assays for HIF-1 target genes Bnip3 and HK1. For each mRNA in each experiment, expression was normalized to the levels in vehicle-treated cells at 20% O_2_. The bars show mean ± SD (*n* = 3 each). (**C**) PC3 cells were exposed to vehicle or the indicated concentration of ALM for 24 h under normoxic or hypoxic conditions and total RNA was subjected to RT-PCR assays for HIF-1 target genes Bnip3 and HK1. For each mRNA in each experiment, expression was normalized to the levels in vehicle-treated cells at 20% O_2_. The bars show mean ± SD (*n* = 3 each). (**D**) Hep3B and PC3 cells were exposed to vehicle or the indicated concentration of ALM for 24 h under normoxic (20% O_2_) or hypoxic (1% O_2_) conditions and cell lysates were subjected to Western blot for HIF-1α and β-actin. (**E**) PC3 cells were exposed to 100 nM of ALM for the indicated time under normoxic or hypoxic conditions and Western blot was performed, * *P* < 0.05 as compared with 20% O_2_, 0 μm ALM group; ^#^
*P* < 0.05 as compared with 1% O_2_, 0 μm ALM group.

**Figure 2 cells-08-00439-f002:**
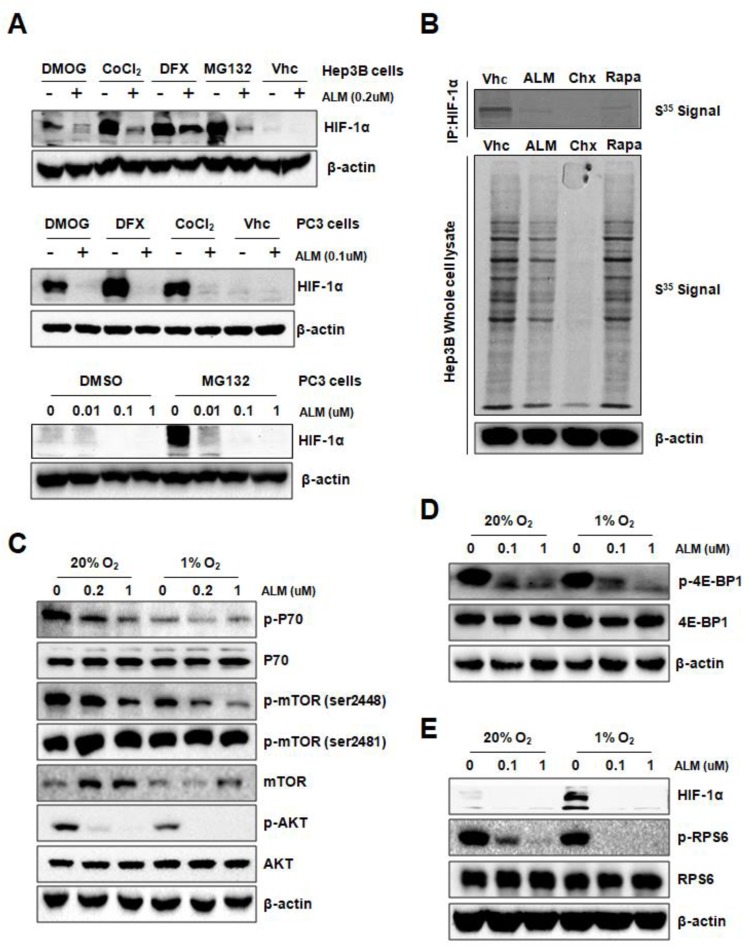
ALM inhibits HIF-1α translation by down-regulating mTOR pathway. (**A**) Hep3B and PC3 cells were exposed to vehicle or the hypoxia mimics dimethyloxalylglycine (DMOG), cobalt chloride (CoCl_2_), desferrioxamine (DFX) or MG132 in the presence of ALM at indicated concentrations or vehicle for 24 h and whole cell lysates were subjected to Western blot. (**B**) Hep3B cells were pretreated for 4 h with vehicle (Vhc), 25 nM rapamycin (Rapa), 20 μg/mL cycloheximide (Chx), or 200 nM ALM, [^35^S] methionine/cysteine was added for 1 h followed by cell lysis, HIF-1α immunoprecipitation (IP), SDS/PAGE, and autoradiography. (**C**–**E**) PC3 cells were cultured at 20% or 1% O_2_ for 24 h in the presence of ALM (at the indicated concentrations) and whole cell lysates were subjected to for: (**C**) p-P70, p-Akt, p-mTOR (2448/2481), mTOR, (**D**) p-4E-BP1, (**E**) p-RPS6, HIF-1α, or β-actin.

**Figure 3 cells-08-00439-f003:**
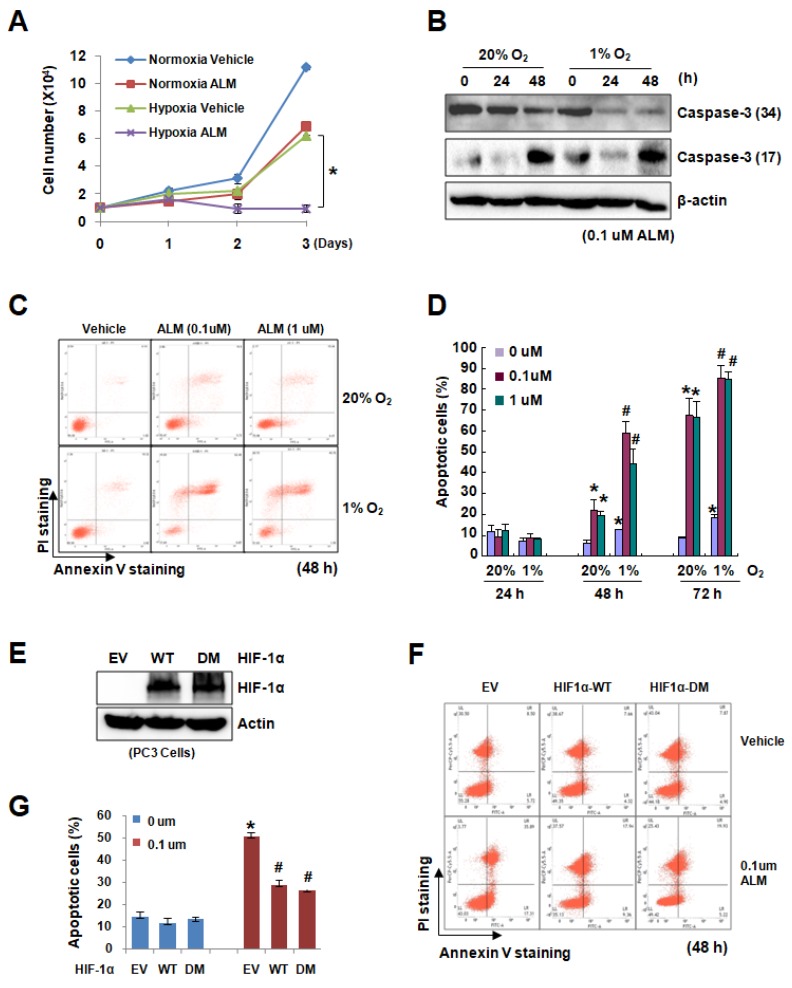
ALM induces HIF-1α-dependent apoptosis in PC3 cells. (**A**) Cell growth curve of PC3 cells exposed to vehicle or 30 nM ALM for 3 days under normoxia and hypoxia. (**B**) PC3 cells were treated with ALM for 24 h and 48 h at 0.1 μM under normoxia and hypoxia, followed by Western blot for caspase-3. (**C**,**D**) PC3 cells were treated with ALM for 24 h, 48 h, 72 h at 0.1 μM and 1 μM under normoxia and hypoxia, followed by staining with FITC-conjugated Annexin V and PI for flow cytometric analysis. Representative result of 48 h was presented as Figure 3C (**C**). The flow cytometry profile represents Annexin V-FITC staining in X axis and PI in Y axis. The number represents the percentages of cells to each of the four quadrants (viable cells in the third quadrant, necrotic or dead cells in the second quadrant, early apoptotic cells in the fourth quadrant and late apoptotic cells in the first quadrant). Then, apoptosis population was quantified (**D**). (**E**) PC cells were infected with lentivirus expressing WT-HIF-1α, DM-HIF-1α (doble mutant, P402A/P564A), and cell lysates were subjected to Western blot for HIF-1α and β-actin. (**F**) PC3 cells stably expressioning EV, WT-HIF-1α, DM-HIF-1α were treated with 0.1 μM ALM for 48 h, followed by staining with FITC-conjugated Annexin V and PI for flow cytometric analysis. The bars show mean ± SD (*n* = 3 each). (**G**) The apoptosis population was calculated based on the flow cytometric analysis of Figure 3F; * *P* < 0.05 as compared with EV, 0 μm ALM group; ^#^
*P* < 0.05 as compared with EV, 0.1 μm ALM group.

**Figure 4 cells-08-00439-f004:**
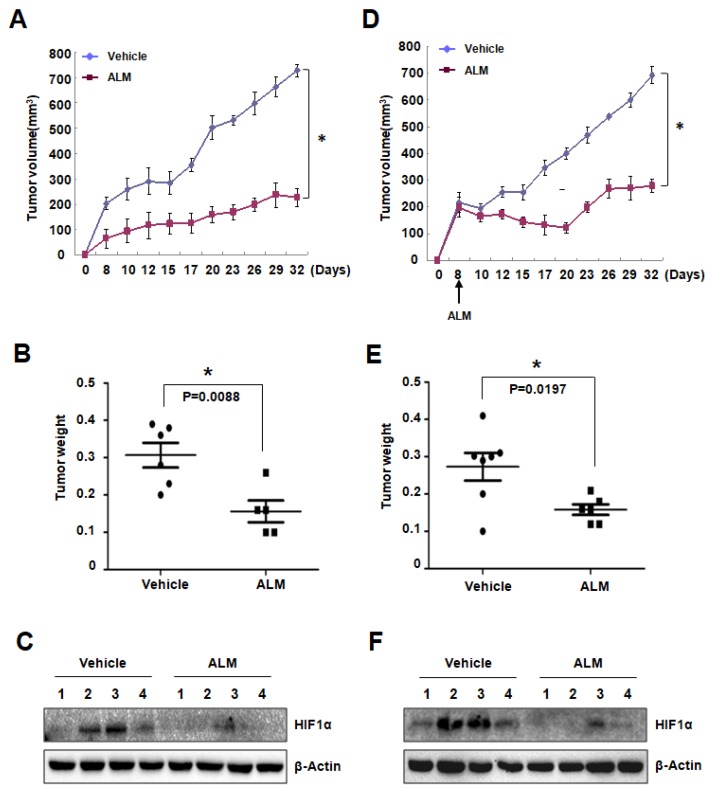
ALM treatment inhibited PC3 xenograft growth. (**A**,**D**) BALB/C Nude male mice were implanted with PC3 cells in subcutaneous tissue on the flank. Mice were treated with daily i.p injection of ALM (10mg/kg body weight, *n* = 5)/vehicle (*n* = 6) starting from 7 days before tumor cell implantation. Tumor volume was determined from day 8 to day 32 (**A**). PC3 cells were subcutaneously (s.c.) injected into BALB/c nude mice. 8 days after tumor inoculation, when the tumor volume reached about 200 mm^3^, mice started to be treated with vehicle (n = 7) or ALM (10 mg/kg body weight, n = 6). Tumor volume was determined from day 8 to day 32 (**D**). mean ± s.e.m. are shown; *P* < 0.0001 vs. vehicle (Student’s *t*-test). (**B**,**E**) Tumors were harvested and weighted after sacrificing mice (mean ± s.e.m.; *P* value is determined by Student’s *t*-test). (**C**,**F**) HIF-1α and protein level was detected in tumor samples; * *P* < 0.05 compared between the indicated groups.

**Figure 5 cells-08-00439-f005:**
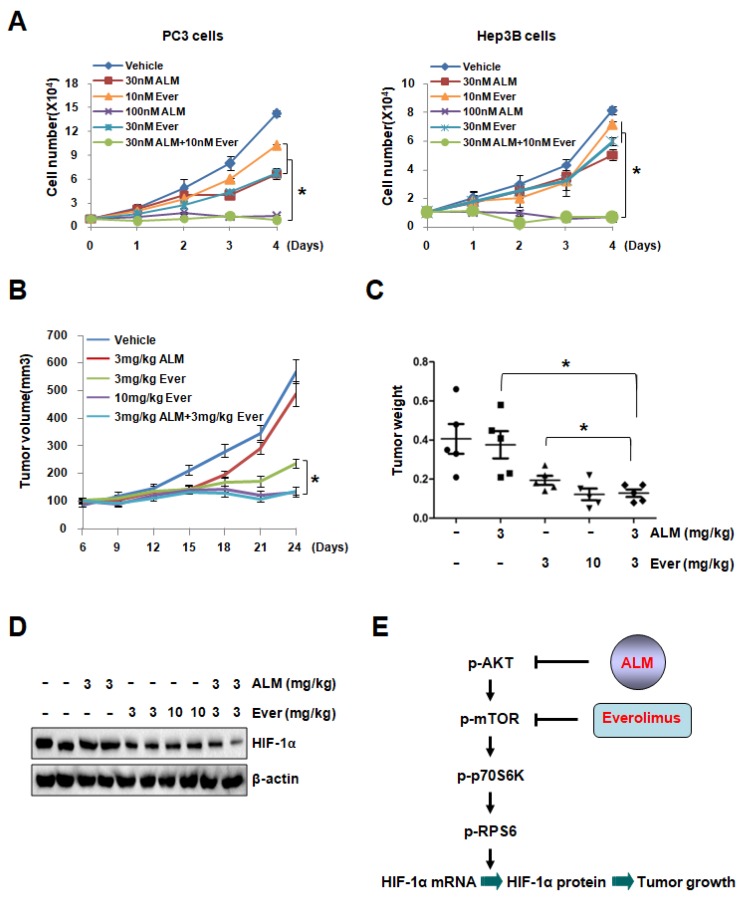
Low dose of ALM enhances the inhibitory effect of mTOR inhibitors on cell growth. (**A**) Cell growth curves of PC3 and Hep3B cells treated with indicated dose of vehicle, ALM, Everolimus, or combination of ALM with Everolimus. (**B**) BALB/C Nude male mice were implanted with PC3 cells subcutaneously on the flank. Mice received daily i.p injection of indicated dosage of Vehicle, ALM, Everolimus or combination of ALM with Everolimus. Tumor volume was determined from day 6 to day 24 (mean ± s.e.m., P < 0.0001 by Student’s *t*-test). (**C**,**D**) Tumors were collected and weighted after sacrificing mice (**C**), followed by Western blot to detect HIF-1α protein level in representative tumor samples of each group (**D**). (**E**) Working model shows that ALM, produced by *Streptomyces flavoretus*, inhibits HIF-1α protein synthesis by down-regulating mTOR signaling cascade. Furthermore, ALM enhances the anti-tumor effect of mTOR inhibitor Everolimus, suggesting its therapeutic potential in the combination therapy for cancer patients, In addition, * *P* < 0.05 compared between the indicated groups, while • indicated Vehicle group, ■ indicated 3 mg/kg ALM group, ▲ indicated 3 mg/kg Ever group, ▼ indicated 10 mg/kg Ever group, and ♦ indicated 3 mg/kg ALM+3 mg/kg Ever group.

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
