# Peer review of "2-Oxonanonoidal Antibiotic Actinolactomycin Inhibits Cancer Progression by Suppressing HIF-1α"

_cells, 2019, doi:10.3390/cells8050439_

Round 1

Reviewer 1 Report

In the manuscript “2-oxonanonoidal antibiotic Actinolactomycin inhibits cancer progression by suppressing HIF-1α”, the authors demonstrate that ALM, an antitumor antibiotic produced by Streptomyces flavoretusy, dramatically inhibited the protein levels in HIF-1α under hypoxia by suppressing its translation via inhibition of mTOR signaling. Further, AML-HIF-1α inhibitory axis resulted in induced cell apoptosis, which in turn slowed down the growth of cancer cells; moreover, the manuscript reports that low dose of ALM could potentiate the anti-tumor effect of Everolimus, an inhibitor of mTOR.

Overall, the work is interesting, carefully written and performed; the rationale for the study is clearly stated but still, there are some issues to be addressed before publication:

-the conclusion that AML-induced apoptosis is HIF-dependent is not fully supported by the available data thus questioning the main conclusion of the manuscript that this is the major mechanism behind the suppressive AML effect on the cancer cell growth. The apoptosis data with cells expressing shRNA against HIF-1α  shown on the Fig.3E-F are supposed to "copy" the effect of AML given the strong HIF-1a inhibition by AML (Fig.1D); those data are rather confusing and contradicting to each other - why cells with HIF-1a knockdown under hypoxia have 30% less apoptosis in the presence of AML that cell in which AML "wipes" HIF-1α protein? How the significant difference in 20% O2 data in the presence of AML without and with knockdown of HIF-1α from Fig.3F-E will be explained in context of AML-HIF-apoptosis axis? Could you provide the levels of the cleaved caspase for the baseline normoxic and hypoxic conditions in PC3 cells?

-there is not visible effect on HIF-1α protein levels in tumors between 3 and 10 mg/kg Everolimus as well as in one of the samples when combination of everolimus and AML were used. Could auyhors elaborate more on that observation?

-it is unusual to show cleaved caspase on different panels (Fig.3B); please provide un-cropped blot

- please explain why the data on Fig 1S showing  60% inhibition of the mRNA levels of HIF-1α in PC3 under hypoxia vs normal oxygen level is not significant?

-how the hypoxia conditions in the study are achieved? The methods and figure legends should be written more accurately to provide the reader with all necessary information for data reproduction and understanding of the statistic. How many biological replicates are done? Why Student test is not applied to all baseline conditions as normoxia and hypoxia?

Author Response

Dear Reviewer,

We are grateful for your comments that well summarized the major findings and significance of our study. Indeed, we discover here that ALM, a metabolite produced by Streptomyces flavoretusy, inhibits tumor proliferation and progression by dramatically repressing HIF-1α translation via mTOR signaling pathway. This discovery is novel and of significance. Meanwhile, we have performed additional experiments and provided explanations to address the reviewers’ concerns and comments. We thank the reviewer for helping us improve the manuscript substantially. Please check the point-by-point response for detailed reply.

Reviewer 2 Report

This manuscript of Cheng et al. describes the effect of Actinolactomycin (ALM) a substance extracted from Streptomyces Flavoretus metabolites as a potential HIF-1α inhibitor by impairing AKT pathway activation and its potential use in future anticancer strategies. Although, the authors efficiently show that Actinolactomycin suppresses HIF-1 expression and activity and that the same substance affects cell proliferation, apoptosis in cultured cells and in mouse xenografts there are still issues to be addressed. 

Moreover, concerning the manuscript itself needs corrections especially in adding details in the methods section (see below) and more imperative in the discussion section where the authors comment mostly on Actinolactomycin's solubility problems and its potential use in combination  with other AKT inhibitors such as Everolimus. However, the main issue of the paper is HIF-1α as indicated by the title and experimentation. So, its imperative that the authors should at least comment a) on the activity of Actinolactomycin under normoxia (in almost all assay) and its implication as a HIF-inhibitor b) how this substance compares with other agents that suppress HIF-1 activity and cell proliferation.

Major points needing to be addressed:

1) Although the authors clearly show that a) ALM inhibits HIF-1α and ΑΚΤ pathway activation and b) on the other hand it affects cancer cell apoptosis, proliferation and tumor growth. However, there is no obvious connection between those observations, since, there are not adequate data to support that HIF-1 is directly implicated in the observed phenotypes exerted by ALM in cancer cells. There is only one experiment (fig 3 E&F) in which the authors try to prove this point but it is not sufficient and its rather confusing. The authors knockdown HIF-1α expression and use in parallel use ALM. Both of these treatments should have the same effect since they both target HIF-1α expression, which A) is not obvious from the figure (according to all known facts knocking down HIF-1 by shRNA should lead to increased death) and, importantly B) is not enough to explain HIF-1 involvement after ALM treatment. One experiment to directly address this point is to force HIF-1α expression (by exogenously expressed HIF-1α or its degradation-resistant mutant) and then treating the cells with ALM. The forced HIF-1α expression should result in increased HIF-1 activity and decreased cell death in the presence of ALM (at least partly).

2) As mentioned above the authors should at least comment on the considerable activity of ALM under normoxic conditions and the implication of this fact in the potential use of ALM as an anticancer agent.

Other points to be addressed :

A) Material and Methods section

1) How hypoxic conditions are achieved? Describe equipment, treatments e.t.c.

2) Provide regent concentrations for used buffers (e.g. RIPA, TBST). Although can be considered common it is correct to include them.

3) It is not mentioned which is the vehicle agent for all experiments. Is it DMSO mentioned in the discussion section? If so, are there any implications for independent researchers to consider? (e.g. how to solubilize agent or treat cells and animals (according to what it is said in the discussion section).

B) Results section

1) Provide references for HK1 and BNIP3 being HIF-1 specific target genes.

2) In figure 1A it is mentioned that the units are RLA (Relative Luciferase Activity) although the graph depiction has a reference point to normoxia which is 1. I think it should be corrected as Fold Increase of RLA in relation to normoxia (both in graph and legend).

3) Figure 1D the HIF-1α panels are out of proportion in relation to actin panels (Seem like they are zoomed in). Please crop and replace with corrected proportions.

4) In all figures the significance levels between normoxia and hypoxia conditions should be shown because ALM activity seems considerable (e.g. graphs in figure 1, 3....)

5) In animal studies representative images of resulting tumors (in the presence or not of ALM) should be included in order to appreciate the result graphs.

C) Supplementary Material

In Figure S3 the graphs and images and text are messed up.

D) In all Manuscript

Please provide uniform and correct symbolism (Onot O2, μg not ug, e.t.c.)

Author Response

Dear Reviewer,

We appreciate greatly the very constructive comments and suggestions from you. Accordingly, we have performed additional new experiments and addressed all your concerns and comments. We thank the reviewer for helping us improve the manuscript substantially and please check the point-by-point response for detailed reply.

Round 2

Reviewer 1 Report

my major concerns are addressed

Reviewer 2 Report

The authors addressed my concerns.

This manuscript is a resubmission of an earlier submission. The following is a list of the peer review reports and author responses from that submission.